# Understanding How Cells Probe the World: A Preliminary Step towards Modeling Cell Behavior?

**DOI:** 10.3390/ijms24032266

**Published:** 2023-01-23

**Authors:** Pierre Bongrand

**Affiliations:** Laboratory Adhesion and Inflammation (LAI), Inserm UMR 1067, Cnrs UMR 7333, Aix-Marseille Université UM 61, 13009 Marseille, France; pierre.bongrand@inserm.fr

**Keywords:** artificial intelligence, biomaterials, cell adhesion, clustering, environmental landscape, mechanotransduction, machine learning, roughness, systems biology, topography

## Abstract

Cell biologists have long aimed at quantitatively modeling cell function. Recently, the outstanding progress of high-throughput measurement methods and data processing tools has made this a realistic goal. The aim of this paper is twofold: First, to suggest that, while much progress has been done in modeling cell states and transitions, current accounts of environmental cues driving these transitions remain insufficient. There is a need to provide an integrated view of the biochemical, topographical and mechanical information processed by cells to take decisions. It might be rewarding in the near future to try to connect cell environmental cues to physiologically relevant outcomes rather than modeling relationships between these cues and internal signaling networks. The second aim of this paper is to review exogenous signals that are sensed by living cells and significantly influence fate decisions. Indeed, in addition to the composition of the surrounding medium, cells are highly sensitive to the properties of neighboring surfaces, including the spatial organization of anchored molecules and substrate mechanical and topographical properties. These properties should thus be included in models of cell behavior. It is also suggested that attempts at cell modeling could strongly benefit from two research lines: (i) trying to decipher the way cells encode the information they retrieve from environment analysis, and (ii) developing more standardized means of assessing the quality of proposed models, as was done in other research domains such as protein structure prediction.

## 1. Introduction: Aim of the Paper

A long-term goal of cell biologists consists of building workable models of cell behavior [1]. The expected benefit is threefold: (1) To predict the fate of cell populations as a function of external conditions. (2) To manipulate cell fate, e.g., to improve patient treatment. (3) To understand the relationship between cell components and function. While this goal has long seemed out of reach of current research endeavors, progress in combining high-throughput data gathering and data processing (e.g., [2,3,4]) warrants the search for specific research strategies to address this problem more directly. As recently emphasized [5], the performance of artificial intelligence and machine learning is dependent on the choice of parameters used to feed currently available sophisticated algorithms. Therefore, an essential requirement for further model development consists of defining as precisely and quantitatively as possible the parameters used by cells to process and encode environmental cues in order to adapt their decisions. In addition, since studies performed on complex processes such as cognition strongly suggest that complex decisions may be based on a restricted set of features selected from an overwhelming amount of information [6,7], it would be attractive to use the dimensional reduction capacity of artificial intelligence [3] to identify a limited set of features used by cells to take decisions. This would provide an invaluable help to our understanding of cell function.

The aim of the present paper is (i) to suggest that currently available machine learning tools might be tentatively used to develop models for prediction of cell response to calibrated stimulation without including the enormous amount of information currently used to connect cell states and signaling networks, including genome, transcriptome, proteome, metabolome and interactome, but with a more precise description of external conditions. (ii) To provide a basis for this strategy by building a workable description of environmental cues used by living cells to take decisions.

## 2. Why the Description of Information Influencing Cell Fate Decisions Is a Timely Question

In order to determine the current bottlenecks hampering approaches to the final goal of whole-cell modeling, it was felt useful to present a brief sketch of data and tools currently available to dissect cell function. This will provide a basis for a discussion of current limitations and challenges.

### 2.1. Current Research Strategies Followed to Decipher Cell Function

#### 2.1.1. Detailed Study of Individual Cell Components

Nearly five decades ago, the development of recombinant DNA technology and monoclonal antibody production gave a new impetus to cell biological research. As stated in a well-known treatise: “Even the minor cell proteins thus became accessible to the most sophisticated structural and functional studies” ([8] pp. 191–192).

In some cases, a definite cellular outcome could be ascribed to the engagement of a single receptor species by a specific ligand, leading to a fairly simplistic view of cell function. As a representative example, soon after the use of monoclonal antibodies to characterize the membrane molecules of blood leukocytes, a leukocyte integrin dubbed LFA-1 (Lymphocyte Function Associated 1) was shown to play a prominent role in leukocyte adhesion to target cells, which was a key step of the immune function. A genetic defect of LFA-1 expression was soon found to cause a specific disease that was called LAD (Leukocyte Adhesion Deficiency, ([9] p. 399). Moreover, definite genes were found to drive autosomic recessive diseases such as cystic fibrosis, which result from a defective anion channel ([9] p. 1986). This was a clear incentive to study the structure, function and intracellular fate of a growing number of molecular species.

#### 2.1.2. Building Exhaustive Datasets: The Omic Enterprise

The remarkable successes of studies of individual molecular species was an incentive to measure all the cell properties that seemed endowed with a functional role, with the unformulated hope that cell function might be understood when all components were characterized at the structural and functional levels. High-throughput methods were accordingly devised and used to study all cell features that were thought to play a significant role in cell function.

Since genetic studies showed that DNA encoded the essential information required to make a living cell, it was warranted to determine the structure of the whole **genome** as a basis for a complete understanding of the function of living cells and organisms. Much effort was thus devoted to determine the genetic structure and variations of a growing number of biological species.

However, since the function of a given cell population results from the selective activation of a specific gene subset, it was rapidly recognized that the **transcriptome** should contain more information than the genome on the workings of an actual cell. This was an incentive to develop powerful methods for RNA sequencing of cell populations and even **individual cells**. Since these methods required the destruction of studied cells, the obtained information appeared as series of snapshots, and suitable extrapolation procedures were needed to build dynamic models of cell function. However, this limitation was recently claimed to have been overcome, which may result in marked advances in the near future [10].

While gene transcription certainly gives a better account of the state of a particular cell than DNA sequence, cell function is essentially driven by proteins and small messenger molecules rather than nucleic acids. However, studying the cell **proteome** is made more difficult by the wide occurrence of protein modification after synthesis. As an example, understanding the signaling status of a cell requires an exhaustive knowledge of the **phosphoproteome**, since phosphorylation events act as molecular switches determining molecular activity. Moreover, protein functional properties, not only concentrations, are key determinants of cell behavior. A prominent example is the set of molecular interactions (the **interactome**). Indeed, it seems well accepted that protein–protein interactions represent the main mode of the proteome function in the cell [5,11,12].

However, even an exhaustive compendium of the concentration and post-translational modifications of all proteins in a given cell would not suffice to account for its state, since this state is also dependent on the localization of molecular constituents. Indeed, cell activity is determined by molecular complexes (such as **signalosomes**) that are determined by the binding properties and co-localization of tens of molecular species. Moreover, the behavior of a cell at a given moment may depend on the time dependence of aforementioned parameters, which led to the defining of parameters such as “RNA velocity”. The importance of the time dependence of cell composition is emphasized by the term of **dynamicome** that was recently coined [13].

Thus, during the last two decades, there was an impressive development of more and more powerful high-throughput measurement methods, resulting in the collection of large and diverse datasets that might enclose a growing part of the information needed to understand cell function. However, even the quantitative determination of the tens of thousands of parameters accounting for the molecular composition and interactions of a single cell would not suffice to model its function in view of two more difficulties that will be discussed in Section 2.1.3 and Section 2.1.4 below.

#### 2.1.3. Achieving a Sufficiently Quantitative Description of Molecular Properties

##### Single Molecule Studies

It was soon apparent that the essentially qualitative data that were provided by omic approaches would not be sufficient to model cell function. As was emphasized in a *Nature* editorial [14], “while molecular biology seemed well on the way to become a largely qualitative science, a **quantitative description** of the properties of cell components is required to understand their function”. Adhesion molecules provide a representative example that is highly relevant to the purpose of this review. As will be detailed in Section 3, a major way for cells to extract information from their environment stems from the interaction of the multiple receptors borne by their highly mobile membrane with specific ligands anchored to surrounding surfaces. These interactions play a major role in decision making. A frequent and important cell decision consists of sticking to a surface bearing molecules recognized by its membrane receptors. Since adhesion is an important process that influences nearly all aspects of cell biology, much effort was made to characterize adhesion receptors. However, an exhaustive listing of the receptors borne by a cell and surrounding surfaces did not allow any quantitative prediction of the outcome of a particular contact between the cell and the surface ([15] p. 227). Indeed, to determine whether two molecules will bind together when they are brought into transient contact, it is necessary to know the rate of bond formation as a function of the distance between surfaces, and once the bond is formed it may be important to know how the retraction force generated by a cell may influence its lifetime [16]. Four decades ago, it was recognized that this information could not be obtained with the conventional framework available to describe molecule interactions [17]. Thus, concomitantly with the omic enterprise, specific methods were developed to quantify interactions between surface-anchored molecules at the single bond level [18]. These advances were used to demonstrate that cell decisions triggered by the interaction between membrane receptors and surface-bound ligands might be strongly dependent on the mechanical properties of these interactions [19].

##### Detailed Study of Small Molecular Assemblies

The very size of omic datasets makes them difficult to interpret. Again, cell adhesion provides a clear-cut illustration: Cell adhesion to solid surfaces currently involves the formation of so-called focal adhesions that act at the same time as force sensors and tethers. However, while extensive omic studies revealed the involvement of nearly 1000 protein species in focal adhesions [20], accurate mechanistic information was obtained by quantitatively studying the behavior of small molecular subsets during the adhesion process. As an example, better understanding of integrin function was provided by detailed structural and functional studies on the role of molecules such as talin and paxillin in connecting integrins to underlying cytoskeletal elements [21].

Thus, in addition to the development of high-throughput measurement methods, there is a need for detailed analysis of small molecular systems in order to identify functionally important parameters. However, it must be recognized that, while performing these studies is certainly required to understand cell mechanisms, even simple problems remain far from being solved. Thus, there is currently no standard set of parameters allowing to fully account for the function of the hundreds of cell membrane adhesion molecules [5]. Moreover, while it is well recognized that mechanical signals strongly influence cell–cell and cell–matrix adhesion, underlying mechanisms remain incompletely understood [22,23]. It may thus be warranted to try and build predictive models of cell function without a complete knowledge of the underlying mechanisms.

#### 2.1.4. Putting Everything Together—The Birth and Growth of Systems Biology

As mentioned above, the extraordinarily high number of molecular species involved in even local and transient phenomena made it difficult to derive mechanistic interpretations from available datasets. Cell adhesion, which was first considered as a fairly simple process, again provides an early representative example. Fifteen years ago, a thorough compilation of published data led to the identification of 156 protein species involved in cell adhesion to extracellular matrix components [24], and this estimate was later confirmed by proximity biotinylation [25]. A few years later, as described in Section 2.1.3, a high-throughput proteomic analysis of focal adhesions revealed the presence of 905 proteins [20]. Other examples may be found in the general process of cell activation. An experimental study taking advantage of mass spectrometry revealed that 227 unique proteins were involved in the early events occurring during the first few minutes and even tens of seconds following receptor-mediated T-cell activation [26]. Deriving understandable mechanisms from these results appeared more difficult, as it is well recognized that cell function cannot be viewed as a sum of additive processes [27]. Rather, this must be compared to a **complex network** of interacting components. As a representative example, when aforementioned advances allowed the characterizing of individual mediators of inflammation that were often denominated as interleukins (suggesting a specific relationship with leukocyte interactions) [28], it soon appeared that a given interleukin acted on several targets (a property called pleiotropy), and different interleukins could act on the same target. In addition, the restriction to leukocytes did not hold. This led to the definition of the so-called cytokine network. An important consequence of this organization is that it makes it difficult to identify the role (or roles) of a given component due to the occurrence of redundancy, an important contributor to the robustness of living organisms.

Thus, an exhaustive knowledge of the structure and function of molecules involved in a given cell process is not sufficient to explain or predict the working of that molecule. Accordingly, the progress of high-throughput measurement methods triggered the adaptation or development of data processing tools that had long been ignored by cell biologists. Since the aim of this review is to present the environmental cues that are retrieved by cells to adapt their behavior and should be fed into theoretical models, it was felt appropriate to present a brief (and necessarily incomplete) summary of theoretical tools currently available to process this information. Three, **not necessarily exclusive**, approaches will be considered: (i) a qualitative display of experimental data, (ii) mechanistic models based on known biological mechanisms, and (iii) purely predictive models based on multivariate statistics and machine learning.

##### Graphical Representation of Cell Organization and Dynamics 

While the results of an omic study may often be summarized as an extensive spreadsheet, experimental reports usually include graphs [29], providing a visual representation of data. Molecules such as adhesion molecules, membrane receptors or kinases are represented as nodes that can be linked by edges representing different kinds of structural or functional interactions, such as binding of scaffold molecules, activation or inhibition of, e.g., an enzyme or a gene. This network representation was early recognized as an important tool of systems biology [30] and remains widely used since its introduction ([24,31,32,33,34]).

Since aforementioned networks usually include several hundred nodes, there is a need for additional explanatory concepts to make them informative. As indicated in an early treatise by a leading author [35], a first purpose of the rapidly growing field of Systems Biology was to allow an intuitive understanding of general principles. Thus, remarkable sets of so-called ‘motives’ [36] formed by small groups of proteins were shown to fulfill so-called **emergent** functions. As an example, a set of three molecules X, Y, and Z, such that X activates both Y and Z, and Y also activates Z through another pathway, was dubbed a **coherent feedforward loop** and it was shown to be able to perform specific functions, such as controlling the delay between activating signals and cell responses. Accordingly, a reasonable way of interpreting a graph consists of isolating motives that are significantly more frequent than might be accounted for by a random arrangement of edges [24]. Note, however, that a general difficulty in interpreting the unexpectedly high occurrence frequency of a given motif is to know whether this is due to a functional mechanism or only to the history of random choices that occurred during evolution.

Another simplifying concept consisted of identifying fairly isolated sub-networks called **modules** that were suggested to fulfill well-defined functions in a fairly autonomous way [37]. This concept is appealing, since module autonomy was suggested to play a key role in the robustness and evolvability of living systems [38,39]. As a consequence, trying to detect modules in experimentally studied networks appears as a usual part of data analysis [32,40,41]. As an example, a thorough integrative proteomic and phosphoproteomic study of T-lymphocyte activation [32] allowed the authors to identify 90 modules on the basis of co-clustering analysis and protein–protein interactions. However, it is not easy to demonstrate that a given pattern in a network actually represents a module. Note in particular that a molecular component may be shared by several modules [37]. Additionally, it may be difficult to determine which degree of autonomy of a set of nodes allows it to be considered as a module. Moreover, specific mechanisms may account for a functional separation such as a timescale difference that is not expected to appear on a graph [42,43]. Further, it may be difficult to identify the function of a motif in a signaling network. Finally, it may be difficult to ascribe a complex function to a subtle property of a given molecule [44]. In conclusion, the possibility to represent cell function as a network of modules rather than a network of molecules, which might help illuminate many important mechanisms, remains a distant goal.

Another representation that met with some success was based on the so-called Waddington metaphor: A cell state may be viewed as a marble rolling on some kind of “energy” hypersurface in a highly multidimensional space. Coordinates may represent the concentrations of each component or the activity of individual genes. As will be indicated in the next section, while this appealing metaphor remains widely used as a starting point of current theoretical modeling, making it quantitative usually relies on sophisticated mathematical tools, such as differential geometry, that are not familiar to most cell biologists [45,46,47].

##### Quantitative Models of Cell Function

While ab initio simulation of a cell as an assembly of atoms ruled by basic physical chemical laws might be considered as a means of modeling its behavior, it was recently emphasized that this approach is, and probably will remain for decades, out of reach of available computer power [1].

A natural way of building a quantitative model of a network consists of using a set of **ordinary differential equations** to model the interaction of nodes connected with demonstrated edges, or links. This approach was early applied on simple biological systems with a well-defined function. As an example, the β-adrenergic control of muscle cell contractility was modeled as a network involving several tens of components linked by 49 equations involving 56 parameters [48]. Parameters were derived from the scientific literature, and validation was performed by comparing experimental and predicted values of data such as the effect of drugs on the activity of enzymes or variations of second messengers such as calcium. Equations were based on generic models such as Michaelis kinetics. A later model included 106 molecular species and 193 reactions. The authors concluded that 109 out of 114 predicted outcomes were validated by published data [49]. As another recent example, this basic formalism was recently used to model the gene regulatory network driving the epithelial–mesenchymal transition, which is an important actor of development and cancer [50]. A recent highly ambitious attempt at processing several independent datasets obtained on *Escherichia coli* included as a basis about 10,000 differential equations and 19,000 parameters [51].

A problem with this approach is that it is usually very difficult to determine the correct values of extensive sets of parameters: they may be unknown, or published values may correspond to heterogeneous experimental conditions. This was an incentive to develop more qualitative approaches such as Boolean models consisting of ascribing discrete states to molecular components, typically active or inactive, and determining the evolution by associating logical functions to each node. Indeed, Boolean networks can be readily built on the basis of published data [52]. Intermediate models were also suggested, e.g., by using logic-based differential equations instead of logical functions [53].

A more visual approach consists of building quantitative models of Waddington’s landscape, in which a cell state is viewed as a point or a vector in a multidimensional space. Coordinates might correspond to the activation state of genes (with a total number on the order of 25,000 in humans). The “energy” is obviously difficult to calculate, and even to define. A reasonable means is to take advantage of Boltzmann’s law to derive a free energy from the density of states that may be determined experimentally [54]. In any case, the basic challenge is to build kinetic equations accounting for the time-dependence of the cell state [47]. Note that some kinetic estimates of differentiation events were cleverly derived from experimental studies of RNA splicing in individual cells [55]. As an illustrative example, a mathematical method was developed to model the regulatory programs underlying the reprogrammation of mouse embryonic fibroblasts into stromal or mesenchymal cells [56]. This allowed the processing of the results of 315,000 single-cell RNA sequencing and predict transcription factors and cytokines affecting cell fate. 

Thus, while much effort is currently being made to build quantitative models of cell signaling and differentiation, obtaining a quantitative picture of cell inner workings remains a distant goal. An important question remains unanswered: due to the high number of models that are currently published, it is important to assess the criteria that may be used to validate them. Indeed, while a number of experimental tests and predictions are presented together with any new model, it would be important to determine the number and stringency of tests that are needed to support further use and efforts to improve all presented approaches. This point will be further discussed in Section 2.2.

##### Use of Multivariate Statistics or Machine Learning to Analyze Cell Organization and Predict State Transitions

During the last decade, it appeared that the need to build complete quantitative models accounting for the enormous complexity of cell signaling machinery could be provisionally bypassed by the use of more and more sophisticated tools referred to as multivariate statistics, machine learning or artificial intelligence [57,58,59]. The basic principles mainly consist of ordering large datasets by finding clusters or patterns without any human help (so-called unsupervised learning) or building predictive rules to derive outcome from input data (supervised learning). Datasets are usually split into a training set, used to build rules, and a test or target set, used to check the validity of these rules. These tools met with considerable success in performing everyday tasks such as image or speech recognition [59]. Recently, they brought a major progress in the derivation of protein conformation from amino acid sequences, by efficiently complementing molecular dynamics [60]. An important benefit is the reduction of the complexity of multidimensional datasets using methods such as principal component analysis or subset selection with so-called shrinkage methods such as ridge or lasso ([57] p61). The following examples illustrate the power of these tools.

HT29 carcinoma cells were treated with nine different combinations of epidermal growth factor (EGF) and insulin to induce apoptosis [61]. The activation kinetics of 19 signaling molecules were determined by performing a series of 13 sequential assays during the first 24 h following stimulation. Tools from multivariate statistics were used to reduce the time series to scalar values (such as decay rate or area under the curve), thus allowing outcomes to be derived from sets of input values. Apoptosis prediction was thus derived from 19 × 13 experimental determinations (with three replicates each) that were summarized as 20 scalar numbers for each stimulation procedure. Predictions were performed with a multivariate regression method.

The cell response to combinations of multiple signals was studied by exposing mouse T lymphocytes to 64 (2^6^) combinations of six different cytokines and studying the production of 10 different proteins (six cytokines and four transcription factors). Principal component analysis (PCA) was used to reduce the output dimensionality from 10 to three (since the first three components accounted for 88% of total variability) and the authors used linear regression techniques to relate stimulation to outcome [62].

Immunologists have long tried to relate the activation pathway followed by T cells to the binding properties of activating antigens. Recently, a combination of a robotic platform and deep learning was used to study the activation of CD8+ T lymphocytes by 24 different antigens [3]. The authors collected single-cell supernatants at 12 time points after the onset of stimulation. They assayed seven cytokines as reporters of activation. A total of 51 experiments yielded 280,000 concentration values. Machine learning was used to extract six patterns of activation dynamics, which should provide a marked help for further modeling of immune responses.

A general conclusion from a large number of studies is that a single homogeneous dataset is not sufficient to allow a full characterization of cell states. It is more and more apparent that the behavior of a given cell is determined by a number of features, including gene transcription, protein content, spatial localization [63,64] and overall cell organization. Thus, a current challenge consists of developing mathematical procedures for quantitative processing of multimodal datasets. As a representative example, single leukocyte transcriptome and expression of 228 membrane proteins were analysed, resulting in the identification of 57 clusters with better separation of previously defined subsets than allowed with transcriptome alone [2].

### 2.2. Current Limitations and Challenges

While the recent progress in modeling cell behavior is highly impressive, it may be useful to delineate current limitations that should shape short-term research strategies.

A first point is that the complexity and versatility of current tools provided by machine learning strongly **increase the required size of training datasets**. Indeed, it is well known that widely different sets of parameter values may yield a satisfactory fit with fairly simple experimental curves with a too versatile model. Interestingly, the efficacy of modern machine learning methods, including support vector machines, random forests or neural networks and conventional statistical methods, such as logistic regression, was compared quantitatively by determining their capacity to effect binary choices on the outcome of pathological situations [65]: it was concluded that recent methods might require up to ten-fold larger datasets than more conventional procedures. A review of 71 published studies led to the conclusion that machine learning methods did not outperform logistic regression [66]. A current way of increasing the amount of information contained in a model might be to combine results of multiple studies. However, this may lead to a combination of data obtained under too different conditions, unless an effort is made to standardize published data, as repeatedly suggested in interactome studies [5].

A second point resulting from the power of modern data processing methods is that when multiple partially correlated parameters are fed into a model, the relationships derived between input parameters might not be causal, thus hampering the possibility to extrapolate conclusions to more diverse conditions. This strongly suggests that the efficiency of machine learning might be increased by suitable incorporation of biological knowledge. This is an incentive to try and understand the inner workings of modern data processing procedures that were rightly compared to blackboxes [67].

A third and related point is that it is well known that a basic requirement for genuine scientific progress is that models be **fully testable**. This principle is well accepted, and published reports usually include several checks of the predictions of presented models. The problem is to determine the required stringency of these checks. In other words, how many predictions must be tested to validate a model of signaling networks including tens or hundreds of parameters? Moreover, it is very difficult to assess the validity of clusters or patterns disclosed by unsupervised learning methods. Indeed, even the definition of a “true cluster” may involve some arbitrary choices. Interestingly, the recent progress of protein structure determination may provide useful guidelines for the assessment of sophisticated methods. Unraveling the complexity of the structure of a protein, which may be defined as the set of coordinates of several thousands of atoms, benefited from the association of different powerful experimental techniques such as X ray crystallography or NMR, and theoretical tools including molecular dynamics and machine learning. Importantly, the continual progress of research strategies was guided by a number of benchmarking methods ([68] pp. 41–54) that demonstrated the importance of building reliable quantitative methods to measure the quality of different models (see [69] for a representative example) and systematic testing of proposed advances, as illustrated by the CASP (critical assessment of structure prediction) enterprise [70]. Similarly, it might be useful to develop quantitative tools for estimating the value of currently proposed models of cell function.

This concern is made more important by present day papers being more and more difficult to assess. Even a careful reviewer cannot fully validate a manuscript, due to a lack of time, competence and included information. Indeed, the difficulty of assessing the validity of a method was recently emphasized to hamper the progress of artificial intelligence [71]. An important point is that most methods involve somewhat arbitrary choices that are often hidden. A simple example is about clustering methods that often rely on the calculation of a distance in a multidimensional space. While Euclidean distance is usually chosen, this is by no means the single possible choice [7]. Moreover, results may be influenced by parameter scaling procedures. Indeed, the lack of information about the details of theoretical models was recognized as a general problem [72].

In **conclusion**, currently available experimental and theoretical methods make it a realistic goal to achieve a detailed description of the state of individual cells and predict transitions triggered by environmental cues. Inner working mechanisms should be obtained in a next step [73]. However, while much effort has been made to achieve a multimodal description of cell states, fewer reports are devoted to a similarly multimodal description of the environmental cues that trigger cell state transition. A major point is that most published reports concern the cell response to a few combinations of soluble bioactive molecules. A better description of cell environmental status should improve the quality of both predictive algorithms and of the assessment of cell regulation models. Thus, it might be rewarding to try to relate the signals sensed by cells probing their environment to final decisions without dissecting signaling mechanisms. A first goal would thus consist of achieving a manageable description of the information retrieved by living cells. Accordingly, in the second part of this paper, we shall review current experimental knowledge allowing us to define the information used by living cells to take decisions. It seems reasonable to expect that the aforementioned tools that were developed to describe the cell state might help us provide a quantitative description of environmental cues.

## 3. Description of Environmental Cues

As explained above, our purpose is not to provide a mechanistic interpretation of the mechanisms used by cells to process extracellular cues, which would be a fairly distant goal, but only to build a minimal list and description of extracellular data that are processed by cells to take decisions, which should provide a firm basis for the building of a manageable “**environmental landscape**”. 

### 3.1. Cell Sensitivity to Soluble Bioactive Molecule

We shall first describe the information provided by the analysis of soluble components of the extracellular milieu, since this was the traditional focus of studies on cell signaling.

#### 3.1.1. Instantaneous Concentration of Active Biomolecules

It was concluded from a bioinformatics study that the human genome encoded about 3700 transmembrane proteins, forming the so-called ‘surfaceome’ [74]. More than 1000 of these proteins might be considered as receptors for soluble mediators, including 779 G-protein receptors (GPCRs). Note that intracellular receptors of membrane-crossing molecules such as steroid hormones should also be considered. There is no doubt that many cell decisions concerning metabolic activity, differentiation, proliferation or migration are heavily dependent on the occupancy of a number of receptors, and it might be tempting to consider this information as an acceptable picture of extracellular signals. Indeed, many attempts at analyzing cell signaling pathways consisted of studying the intracellular changes triggered by an exposure to suitably chosen concentrations of ligands of well-characterized cell receptors ([3,26,61,75] provide but a few examples). However, while it might be found more straightforward to relate cell outcomes to receptor occupancy rather than to the composition of extracellular spaces, it must be kept in mind that a given receptor may generate different responses after engagement with different ligands, a phenomenon sometimes referred to as receptor bias. As a recent example, it was shown that different ligands of angiotensin II type 1 receptor could trigger different responses and conformational changes of this (GPCR) receptor [76]. Moreover, as mentioned below, different ligands of a given T-cell receptor (TCR) may trigger widely different responses, from paralysis to full activation, and differences have been mostly ascribed to lifetime differences of TCR-ligand complexes [77,78].

Another important point is that cell responses to several stimuli are not additive. This was well studied in a systematic study of the stimulation of macrophage-like RAW264 cells by 22 receptor ligands (such as growth factors and/or cytokines) that were added either separately or as 231 binary combinations. A large number of non-additive interactions were evidenced [27]. Thus, a quantitative description of cell response to receptor ligands may require studying a large set of ligand combinations. The multiplicity of mediators contributing T follicular helper cell differentiation is but a single example [79].

In conclusion, a detailed list of the concentrations of cell receptor ligands in the extracellular medium is an important component of a list of exogenous cues that determine fate decisions. However, as will be shown in the next section, the temporal variations of these concentrations must also be considered.

#### 3.1.2. Time Dependence of Receptor Engagement

It has long been shown that the outcome of cell stimulation after activation of a given signaling pathway could be related to the **dynamics of signaling events** [42,75,80]. As an early example, the stimulation of the well-known mitogen-activated protein kinase (MAPK) pathway was monitored in PC12 mammalian cells. Epidermal growth factor (EGF) was found to induce cell proliferation, together with a transient MAPK activation, while nerve growth factor (NGF) induced durable MAPK activation and differentiation including neurite outgrowth [42]. It must be emphasized that this finding may seem to contradict the hope that an analysis of cell-signaling networks might be facilitated by the existence of independent modules that might fulfill separate functions [39]. As another early example, it was reported that a transient rise of intracellular calcium induced in B lymphocytes induced NFκB activation, while a lower and more durable calcium rise stimulated the NFAT pathway [80]. Moreover, as mentioned above, different T-cell receptor ligands may induce widely different responses, and this seems tightly related to the kinetics of ligand–receptor engagement (as reviewed in [81]). The importance of the dynamics of signaling phenomena might provide a basis for the so-called **proof-reading** mechanism that ensures the fidelity of intracellular biochemical processes [82,83,84]. 

Note that the relationship between the temporal variation of mediator concentration and the dynamics of receptor engagement and second-messenger generation may not be straightforward [85]. Thus, many reports supported the concept of **dynamical encoding** in cells processing exogenous cues [86]. As an example yielded by a quantitative analysis of the induction of intracellular calcium changes by extracellular ATP, the authors were surprised to find that cell decoding mechanisms acted as low-pass filters, making cell-retrieved information somewhat insensitive to the underlying signal dynamics [87]. This point is highly relevant to our purpose. Indeed, if signal encoding is a general component of intracellular information processing, it may be difficult to relate physiological cell outcomes to specific molecular species. This may match the need to understand logic operations rather than the activity of individual electronic components when studying modern computer functions [37]. 

In conclusion, the aforementioned data show that a quantitative description of extracellular signals requires substantial information on the time dependence of receptor engagement. It is thus important to determine the temporal length and resolution that must be considered, or, in other words, the details of **cell memory** of past events. Here are a few examples: TCR-mediated T cell activation is a well-studied process that seems to be triggered by a summation of binding events during a period of time on the order of 1 min [19,88,89]. Gradient-directed cell migration plays an important role in development and in pathogen elimination. **Directional persistence** results from a balance between directional memory and capacity to respond to rapidly fluctuating chemoattractant concentration, and directional memory was estimated at a few minutes [90,91]. Interestingly, the requirement to account for a memory may set a constraint on theoretical models of cell state, as discussed in Section 2. It has been repeatedly found that cells keep a memory of the mechanical properties of their environment. The duration is on the order of days to weeks, and a theoretical model was recently elaborated [92]. A rather extreme example is provided by the immune system, in which interaction with a foreign pathogen usually results in the generation of so-called memory cells that will provide lifelong protection.

In conclusion, the input of a model of cell response to extracellular cues should include quantitative information on the **time dependence** of the concentration of extracellular ligands. The time scale of the required information is obviously dependent on studied output, and cell response to chemoattractants may depend on second- or minute-scale variations of stimuli. Cell differentiation will rather require time scales of several days to months.

#### 3.1.3. Spatial Heterogeneity of the Extracellular Milieu

In some cases, cell response to an extracellular ligand is determined not only by its concentration and time dependence but also by **spatial distribution**. Chemotaxis provides a clear illustration. As it has long been shown [93], when a motile cell such as a vertebrate leukocyte is exposed to a spatial gradient of a chemoattractant, it rapidly starts migrating along the gradient direction. It was estimated that a relative concentration difference of the order of 1% between two cell sides could be detected. A point of caution relative to the interpretation of this phenomenon may be added: the capacity of a cell to migrate along a concentration gradient does not provide a formal proof that spatial information is retrieved. Thus, bacterial chemotaxis was found to be based on serial measurements of chemoattractant concentration: swimming bacteria usually display repeated random directional changes denominated as tumbling. The tumbling frequency was shown to decrease when local chemoattractant concentration increased, which was sufficient to provide directed migration. In contrast, mammalian cells may process vectorial information [90,93].

In conclusion, the effect of **extracellular medium** on cell behavior might be satisfactorily accounted for by the spatial and temporal distribution of biologically active molecules. 

However, compelling evidence has shown that cell behavior is at least as deeply driven by the properties of **solid structures,** such as an extracellular matrix or neighboring cells, as by soluble molecules. As will be shown below, the mechanisms of information retrieval and the nature of parameters processed by cells encountering outer surfaces may be quite different from those mentioned above.

### 3.2. Cell Sensitivity to Surface Properties of Their Environment

While little was known four decades ago about cell communication via the direct interaction of molecules with the outer surfaces of cells ([8] p. 717), a compelling amount of evidence now shows that cell behavior is highly dependent on multiple interactions with neighboring surfaces. Anchorage dependence is a well-known example: most cells (but not leukocytes) need to adhere to outer surfaces to proliferate and even to survive [94]. Moreover, adhesive interactions are well known to drive prominent cell processes such as spreading [95], migration [96,97], and differentiation [98].

Importantly, while cell response to soluble molecules is essentially determined by **intrinsic** properties of interacting membrane receptors and soluble ligands, the information retrieved by cells interacting with surface-bound receptor ligands is strongly dependent on a number of properties of the **environment** of interacting molecules. The purpose of this section is to provide a list of these properties that have to be entered as an input to any model of cell decision. While a full discussion of underlying mechanisms, which remain incompletely understood, would not fall into the scope of this review, main hypotheses will be briefly mentioned to make this listing more understandable.

#### 3.2.1. Differences between Cell Receptor Interaction with Soluble and Anchored Ligands

It has been amply demonstrated that a given molecule may induce widely different cell responses in soluble and surface-anchored form. Thus, surface-bound but not soluble vitronectin was found to support endothelial cell survival [99]. Moreover, low concentrations of a monomeric TCR ligand were found to activate T lymphocytes only when they were bound to surfaces such as plastic or lipid bilayers [100]. We shall now consider three non-exclusive features that might be responsible for these differences: kinetics of receptor engagement, force generation and ligand topography. In Section 3.2.2, we shall discuss general material properties that were also shown to influence cell behavior under conditions where interacting molecules are not fully characterized.

##### Kinetics of Receptor Engagement 

As indicated in Section 3.1.2, the signal triggered by the exposure of a membrane receptor to a specific ligand is dependent on the number and duration of binding events. The kinetics of soluble ligand binding is determined by the conventional association and dissociation constants that are considered as intrinsic properties of any ligand–receptor couple. In contrast, the binding kinetics of surface-anchored molecules strongly depend on many different parameters, as explained below.

##### Cell-Specific Parameters

While the contacts between cell membrane receptors and soluble ligands are driven by thermal diffusion, the association with and dissociation from surface-bound ligands are triggered by cell movements. Indeed, the cell surface is highly motile [101] and studded with a number of protrusions of varying shape and denomination, such as lamellipodia, invadopodia, filopodia, microspikes or microvilli. Microvilli have a typical diameter of the order of 0.1 µm and a length varying between a few tenths of a micrometer and several micrometers; they have long been considered as exploratory structures, since their tip may be enriched with cell receptors and signaling molecules [102] and they display continual protrusion–retraction cycles. The interaction of microvilli with surfaces have been studied quantitatively with microscopical techniques such as interference reflection microscopy (IRM) or total internal reflection fluorescence microscopy (TIRF). Thus, when myeloid [103] or lymphoid cells [88,104] were sedimented on ligand-bearing surfaces, cell adhesion [103] or activation [88,104] was preceded by the occurrence of transient contacts of a few seconds duration between the tips of microvilli and surfaces. This phenomenon lasting several tens of seconds was dubbed “tiptoeing” [103,105]. 

Obviously, the time available for molecular contact formation between binding sites is dependent on the period of time during which the tip of a microvillus is at binding distance of a surface-anchored ligand. Importantly, this may be influenced by surrounding molecules. Indeed, cell membranes are usually coated with a dense polysaccharide-rich layer called the glycocalyx that may generate repulsive forces and hamper molecular interactions [15,106]. Interestingly, this inhibition may be modulated by active cell processes. Thus, some bulky glycocalyx elements were reported to be sorted out of contact zones through an incompletely understood active mechanism [107]. Moreover, cells such as macrophages were found to increase the avidity of membrane receptors by partial pruning of glycocalyx elements [108]. More recently, the repulsive properties of endothelial cell glycocalyx were measured by indentation of glass microbeads with an atomic force microscope [109]: the glycocalyx behaved as an elastic layer of 110 nm thickness and an elastic modulus on the order of 5 kPa. The authors estimated that this repulsive layer triggered a ten-fold reduction of the number of receptor–ligand interactions between flowing blood granulocytes and the endothelial cell walls.

Another point of importance is the receptor concentration on the tip of microvilli. This was shown to be a key factor of the activity of receptors such as L-selectin [110] or T-cell receptors [111].

##### Molecular 2D Binding Parameters

While the frequency and duration of membrane receptor associations with soluble ligands, under so-called 3D conditions, are fully determined by molecular concentration and conventional association and dissociation rates, a different theoretical framework is required to account for bond formation between surface-anchored molecules, i.e., under so-called 2D conditions [112]. Key concepts will be rapidly summarized below. The reader is referred to a recent review for more information and references [5].

*Bond formation*. Bond formation between a (free) cell membrane receptor and a soluble ligand L is usually viewed as a simple stochastic phenomenon occurring with a probability of k_on_.(L).dt during a small period of time, dt. The on-rate k_on_ provides a global account of the rate of molecular encounters and the probability that a molecular encounter will result in bond formation. This constant is expressed in M^−1^.s^−1^ and may be considered as an **intrinsic** property of the ligand–receptor couple, liable to quantitative determination with standard physical-chemical tools. In contrast, the rate of bond formation between two molecules anchored at distance d, i.e. under so-called 2D conditions, has a dimension of second^−1^, and it is not a **number** but a **function** k_on_(d) [112]. This function is not easy to determine, and it cannot be considered as an intrinsic property of the receptor and ligand binding sites, since it is dependent on the **length** of interacting molecules (usually several tens of nanometers). Molecular shape and **flexibility** are also important to allow contact between binding sites. As an example, integrin molecules may display an inactive form associated with so-called closed conformation and reduced distance between the membrane and binding site [113,114]. Moreover, the rate of encounters between binding sites is dependent on the lateral diffusion of receptor and ligand molecules, which is determined by cellular properties. In conclusion, the capacity of a membrane receptor to interact with a surface-anchored ligand is dependent on cell motion, cell surface topography and molecular composition, and also on binding parameters that are not included in the conventional framework used to describe 3D interactions.

*Bond lifetime*. The rupture of a bond formed between a cell membrane receptor and a soluble ligand may be viewed as a stochastic event with occurrence probability k_off_.dt during a small time interval dt. The off-rate k_off_ is a constant number expressed in second^−1^ and may be considered as an intrinsic property of the ligand–receptor couple (provided the complex is exposed to a standard temperature and medium composition). However, the lifetime of a bond formed between membrane-anchored molecules is determined by a balance between the force F generated by the **cell-driven retraction** process and the bond mechanical **resistance**. The force dependence of the dissociation rate of biomolecular bonds has been subjected to intensive study during more than three decades. It was first thought that k_off_(F) followed a simple exponential law (often referred to as Bell’s law [115]). It was later found that the rupture process might display a more complex behavior: so-called **catch bonds** were found to display increased lifetimes in the presence of forces in the order of several piconewtons. Moreover, it was found that bond formation behaved as a multistep process with a progressive increase of mechanical strength. The dissociation rate is thus a function k_off_(F,t), where F is the applied force and t is the bond age. A last point is that rebinding of two anchored molecules maintained at close distance may conversely increase the total duration of receptor engagement and thereby enhance signaling [116].

In conclusion, the frequency and duration of binding events between surface anchored molecule is strongly dependent on cellular properties.

##### Force Generation

During the last two decades, a wealth of experimental data demonstrated that force generation between anchored ligands and membrane receptors plays a major role in signal generation. Indeed, it is well established that (i) forces exerted on membrane molecules can trigger signals, (ii) forces are generated at the interface between cells and surfaces, and (iii) the tethering strength of anchored ligands actually influences cell activation. We shall describe a few examples to illustrate these points. More information and references may be found in a recent review [117]. However, for the sake of clarity, it was felt useful to recall the precise meaning of the concept of signal used in this and our previous [117] paper. It was deemed reasonable to define a signal as a message that may be generated by any interaction between a cell and its environment, such as the engagement of a membrane receptor by a specific ligand molecule. Living cells are exposed to countless signals that may induce responses. A cell response often consists of so-called signaling cascades, i.e., sequences of intracellular biochemical events resulting in changes of cell state and behavior.

*Piconewton forces may generate signals*. There is no doubt that a major mechanism of signal generation following receptor engagement is the induction of a receptor conformational change by the binding process. The aforementioned G-protein coupled receptors (GPCRs) probably provide the best studied example: the engagement of transmembrane GPCRs will allow them to interact with heterotrimeric Gαβγ proteins and trigger a signaling cascade. It is now well demonstrated that piconewton forces may be sufficient to trigger a conformation change and disclose binding sites, leading to signalosome assembly. Thus, forces of the order of 50 pN [118] and even 10 pN [19] applied on T-cell receptors were found to trigger cell activation. Forces as low as a few piconewtons were found to change the conformation of talin, a protein found in focal adhesions with a role in connecting integrins to actin microfilaments [44,119]. Piconewton-scale forces were shown to activate membrane cationic PIEZO channels [120].

*Cells are known to generate forces of several tens of piconewtons in contact with outer surfaces*. Granulocytes deposited on fibronectin-coated pillars of 500 nm diameter exerted protrusive and retraction forces of several tens of pN/post [121]. T lymphocytes generated traction forces of the order of 100 pN on arrays coated with antibodies to CD3, a TCR-associated membrane structure [122]. Integrin-mediated adhesion of melanoma cells to surfaces involved traction forces of at least 40 pN [123].

*The capacity of surface-anchored receptor ligands to activate cells may require a minimal tethering strength*. Indeed, the adhesion of CHO cells to the integrin ligand RGD required that tethers connecting RGD motives to surfaces resist forces higher than 40 pN. However, a tethering strength of 12 pN was sufficient to allow the activation of the Notch receptor by surface-anchored ligands [124]. Moreover, the initiation of antibody production by B lymphocytes stimulated with antigen-anchored surfaces was influenced by the strength of antigen-surface bonds [125].

Thus, the simple concept suggested by the aforementioned experiments is that cells analyze surrounding surfaces by forming multiple intermolecular bonds and subjecting these bonds to forces generated by multiple protrusion-retraction cycles. Interestingly, a theoretical analysis showed that force application on receptor-ligand bonds allowed more rapid and/or precise analysis of the interaction properties than a mere determination of receptor occupancy by soluble ligands [126].

##### Spatial Distribution of Ligands on Surfaces

Molecular clustering is an important mechanism of signal generation. For example, this may allow a kinase to encounter and phosphorylate a specific site on a target molecule. Thus, the activation of an EGF receptor by ligand binding results in dimerization and autophosphorylation. An early step of T lymphocyte activation is the phosphorylation of specific sites called ITAMs on the TCR complex by kinases such as lck [81]. While clustering may be triggered by a conformational change induced by monovalent ligand binding, multivalent receptor ligands often behave as powerful clustering triggers. Thus, T lymphocytes are efficiently activated by antibodies specific for the complex made with TCR and CD3 on their membrane. Indeed, this activation has long been used as a reporter of T lymphocyte function to help in the diagnosis of immune deficiency diseases. Further, it has been emphasized that the communication of different cell types such as immune or neural cells may involve the formation of specific molecular assemblies called synapses in contact zones, and the potential role of these synapses in regulating signal transduction has been studied for decades [127,128]. Clearly, a specific pattern of binding molecules exposed on a surface may influence the organization of membrane molecules in a cell adhering to this surface.

Accordingly, the outcome of cell interaction with anchored receptor ligands may be highly dependent on their spatial distribution. An early and impressive example is the demonstration that endothelial cells were switched from growth to apoptosis when they were deposited on surfaces bearing adhesive islands of decreasing size, from hundreds to tens of squared micrometers [129]. A possible mechanism might be that a minimal adhesive area is required to allow cell spreading [130], involving increased microfilament polymerization and apparent area increase, together with volume and thickness decrease and marked metabolic changes. More recently, it was found that cells were also highly sensitive to the nanometer-scale topography of anchored ligands. Thus, rat fibroblasts displayed efficient spreading on surfaces coated with integrin ligands organized in hexagonal patterns separated by 58 nm intervals, not 110 nm intervals [131]. In addition to the distance between ligand molecules, their geometrical ordering was also found important: disordering the surface distribution of integrin ligands on a surface could strongly increase the adhesion of breast myoepithelial cells [132]. Further, much experimental evidence showed that different regions of a given cell may react differently with a same ligand. Thus, an intercellular contact was reported to generate an attraction on the rear side of a cell, and a repulsion on the anterior region of the same cell [133]. 

Another important point is that the outcome of cell interaction with ligand-coated surfaces is not only influenced by the lateral (2-dimensional) distribution of membrane molecules, but also by the width of the separation gap. This width is determined by the length of adhesion molecules and the glycocalyx-generated repulsion, and it may display wide variations, from tens to hundreds of nanometers [134,135]. As a consequence, bulky molecules may be sorted out of the contact zone and replaced with smaller molecules. A striking illustration of the potential importance of this mechanism is provided by the exclusion of the bulky CD45 phosphatase (about 45 nm width) from the region of contact between T lymphocytes and antigen-presenting cells. It was predicted that this exclusion might trigger T-lymphocyte activation by facilitating the phosphorylation of specific tyrosine residues [136]. This was checked experimentally by showing that an elongation of the molecule bearing a TCR ligand reduced its capacity to activate T lymphocytes [137] and that phosphatase exclusion might suffice to trigger cell activation in a model system [138]. More recent experimental evidence supports the concept that local separation distance of interacting surfaces can influence signaling [139]. 

In conclusion, in addition to their sensitivity to soluble mediators, cells are strongly influenced by the nature, density and topography of active biomolecules exposed to surrounding surfaces, including extracellular matrix (ECM) and neighboring cells. Further, the effect of a given molecule on cell fate decisions may be different when it is in soluble form or anchored to a surface. This information should thus be included in the input of any cell model.

In addition to the properties of anchored bioactive molecules, cells are known to be strongly influenced by the material properties of their microenvironment [140]. We shall now list the features of recognized importance. Note that the separation between the material properties of anchored molecules and underlying substrates may seem somewhat arbitrary, but this was felt warranted for the sake of clarity.

#### 3.2.2. Influence of Surrounding Material Properties on Cell Behavior

It is well known that the properties of an extracellular matrix or neighboring cells play an essential role in cell behavior. Anchorage dependence, which was mentioned above, provides a dramatic example [94]. However, it was long considered that this phenomenon was essentially due to the necessity for cells to interact with specific anchored molecules. More recently, it was repeatedly demonstrated that cells are also deeply influenced by some bulk properties of surrounding surfaces, including basic physical-chemical properties such as hydrophobicity, mechanical stiffness, and micrometer- or nanometer-scale topography. Properties probed by cells encountering a surface will now be considered sequentially.

##### Basic Physical-Chemical Properties

When the mechanisms of cell interaction with other cells or surfaces were first studied, it was hoped that the basic theoretical framework developed by physical chemists [15] might provide a strong basis for data interpretation: a suitable Hamaker constant might account for an attraction between membrane lipid bilayers. The negatively charged glycoconjugate-rich membrane coat, called the glycocalyx, might account for an anti-adhesive effect generated by electrostatic repulsion and entropic repulsion generated by the confinement of flexible polymer, a phenomenon called steric stabilization. The capacity of phagocytic cells such as macrophages to ingest hydrophobic particles might be accounted for by surface-energy effects (see [15] for a general presentation and [141] for more information and specific references). Indeed, an important issue in biomaterials science would be to find a set of parameters allowing researchers to fully characterize a surface and predict its fate in a biological environment [142]. However, this hope has not yet been fulfilled. While the importance of basic parameters such as surface charge and hydrophobicity remains recognized in current thinking [143], the parameters provide insufficient information to predict the outcome of cell–cell and cell–surface interaction. An important reason for this situation is that a material surface exposed to a biological environment is coated within seconds with a layer of adsorbed biomolecules that subsequently undergo conformation changes that will result in the exposure of a number of structural motives, the interaction of which with the multiple cell membrane receptors will drive biological responses [141,143]. Moreover, while glycocalyx-mediated repulsive forces are certainly important, their effect may depend on considered receptors. Thus, long adhesion receptors such as P-selectin (about 40 nm length) are much less sensitive to repulsive forces than T-cell receptors or CD32 immunoglobulin receptors (about 15 nm length). Thus, the information sensed by a cell encountering a surface is better described by a description of the accessibility of individual ligand molecules than a global parameter accounting for repulsion and surface separation.

##### Environmental Stiffness

It has been repeatedly demonstrated that many cell processes such as resisting mechanical forces [144], migrating, [145,146], spreading [147], entering an activation program [148] or differentiating [140,149,150,151] are highly dependent on the stiffness of underlying surfaces. Therefore, it is important to define an exhaustive set of parameters accounting for the mechanical properties of surfaces that are sensed by cells. In most cases, experiments were performed with polymers such as polyacrylamide [149,151], collagen [150] or alginate [140] coated with adhesion molecules such as fibronectin. Young modulus is commonly used as a reporter of surface stiffness. This parameter provides a simple way of describing the elasticity of a homogeneous medium, i.e., the deformation induced by a constant stress. However, it is not ensured that this provides an exhaustive account of the information retrieved by a cell probing an actual surface. Firstly, actual media are viscoelastic, and it is important to consider the timescale of stiffness determination by cells [92,152]. Indeed, when mesenchymatous cells were deposited on hydrogels exhibiting independent changes of elasticity and stress relaxation, spreading, proliferation and differentiation were shown to be influenced by stress relaxation parameters [153]. Murine fibroblasts and osteoblasts were reported to detect substrate prestrain [154], and the spreading of mesenchymal stem cells was concluded to depend on matrix plasticity [155]. It would certainly be useful to determine a small set of parameters accounting for the mechanical properties of most substrates encountered by cells under experimental or physiological conditions. Moreover, since the major path followed by cells to probe surface stiffness is probably by pulling at ligands of adhesion receptors, retrieved information is expected to depend at the same time on the mechanical properties of the substrate, adhesive bonds, receptor-bearing molecules and receptor–cytoskeletal interaction. Thus, a ligand-specific tethering parameter might provide a better description of the information yielded by a given adhesive interaction. Accordingly, cell stiffness sensitivity is dependent on the arrangement of adhesion sites [156], and it was concluded from other studies that cells integrate adhesive and mechanical cues provided by a substrate [157,158].

Probably different sensing mechanisms are used by cells to process **pulling** and **pushing** interactions. Indeed, traction forces may be used to assess the biological significance of a ligand or the opportunity to develop focal adhesions, whereas a pushing movement may generate inhibitory signals to a moving cell [159]. Specific molecules may be needed to inhibit cell growth, since contact inhibition is required to maintain the homeostasis of an entire organism. It is also important for a migrating cell to be able to stop pushing against a stiff surface. Therefore, the possible importance of using different parameters to account for substrate response to pushing and pulling forces should be considered [133].

##### Surface Topography

It has long been known that adherent cells are highly sensitive to the micrometer- and nanometer-scale topography of underlying surfaces [160].

Thus, cells align along grooves exposed by a surface [161,162]. The amoeboid migration of lymphocytes confined within channels of a 5 µm section was sensitive to the micrometer-scale texture of these channels [163]. The migration of breast cancer cell deposited on fibronectin-coated surfaces was sensitive to the micrometer-scale shape of adhesive areas [164].

Further, cell behavior is markedly sensitive to the nanometer-scale topography of surfaces. As an example, macrophage activation was qualitatively dependent on the roughness of underlying titanium surfaces [165,166]. When T lymphocytes were deposited on surfaces bearing nanopillars of 10 nm height bearing anti-TCR antibodies, activation was decreased when nanopillar spacing was increased to above 50 nm [167].

A problem to make use of these results is that there is no simple way of achieving an exhaustive description of surface roughness [168]. A possible way of identifying a minimal set of parameters accounting for the influence of environment roughness on cell function might be to consider potential mechanisms for this influence. The following three hypotheses may be considered: (i) As mentioned above, the signaling efficiency of a membrane receptor may be modulated by its molecular environment. As described above the sorting out of phosphatase-associated bulky CD45 molecules was shown to be involved in TCR-mediated signaling. This sorting was altered when TCR ligands were bound to nanopillars with a spacing wide enough to allow CD45 insertion [167]. (ii) It has been well shown that the membrane of cells strongly adhering to a surface may at least partially adapt the cell’s curvature to improve molecular contact [134,169]. Further, the conformation [170] and localization of membrane proteins may be influenced by local curvature, thus modulating signaling. Recently, cells expressing a fluorescent actin reporter were deposited on substrates bearing nanostructures of varying curvature [171]; the authors concluded that actin fibers formed in regions of curvature radius lower than about 200 nm, and this involved FBP17, a curvature-sensing protein. (iii) During the first phase of contact formation between a cell and a surface, the formation of ligand–receptor contacts may be strongly modulated by the topography of approaching surfaces [172], leading to the concept of an effective contact area [173]. Alternatively, it might be useful to define an effective density to account for the accessibility of any important ligand molecule.

A tentative summary of parameters sensed by living cells to analyze their environment is shown in Table 1.

## 4. Conclusions

The main conclusion is that while much progress has been made in providing quantitative accounts of cell states with thousands of parameters, and in analyzing the state transitions triggered by tens of combinations of soluble stimuli, much experimental evidence has demonstrated that cell behavior is guided by a complex combination of multiple environmental cues, including the density of soluble and anchored molecules, 2D and 3D topography of the surrounding matrix and cells at the micrometer and submicrometer level, and the mechanical properties of these components. It is therefore suggested that it might be worthwhile to try to build a quantitative description of the cell environment with an accuracy matching the currently achieved description of cell states. It might be rewarding to relate detailed environmental signals to physiological outcomes without including information relative to intracellular networks. It is suggested that this might provide a valuable support for the modeling of cell state transitions. As a first and essential step to approach this goal, there is a need to build sufficiently extensive training datasets, including a quantitative account of the aforementioned signals and cell behavioral responses.

In addition, it was felt that the following three remarks might be of interest.

First, it might be warranted to improve the tools used to check currently developed models. Indeed, while innovative computer-assisted tools allow processing of the enormous amount of information yielded by high-throughput omic studies, a major weakness caused by the very power of these techniques is that it is difficult to subject analyses to stringent checks. Unsupervised clustering or principal component analysis are representative examples of heuristic tools yielding results that are difficult to validate. This concern is the more serious, as the growing complexity and multidisciplinary character of systems biology makes the details of current papers more and more difficult to criticize, and even to understand by reviewers with a biological background. Interestingly, the importance and dangers of these situations have been encountered and well stated in other scientific domains, such as the highly respected field of theoretical physics [174,175,176,177]. Thus, it would certainly be useful to organize systematic checks of cell behavioral models. The support of CASP experiments to the development of protein structure predictions suggests that this may be a rewarding effort [70].

A second point is about the data efficiency of elaborated models. Current omic-based descriptions of cell states may involve hundreds or thousands of parameters. Building a readily testable model of signaling processes would require a drastic dimensional reduction. A current challenge certainly consists of combining conventional biological wisdom with machine learning to improve the efficiency of data processing. This strategy was claimed to be successful in the domain of image processing [178]. Moreover, the human brain is known to be able to rapidly analyze an image that may include millions of pixels, e.g., for face recognition that has to be completed fairly rapidly. This can be achieved by using a specific encoding [6,179]. Similarly, there may be a need for cell survival to synthesize a myriad of environmental cues into a simple and fairly stable class. It is tempting to speculate that it might be possible to identify a cell code for environmental description.

Third, as suggested in the beginning of this section, the quality of training datasets should determine the success of future progress in modeling cell behavior. A first and easiest step consists of listing relevant parameters, in the spirit of Table 1. However, the second and much more formidable task will consist of filling the cases of training spreadsheets with relevant information on the effect not only of individual signals but also of signal association (see, e.g., Section 3.1.1 and [27]). This will obviously require a much more extensive and up-to-date analysis of published data than the mere compilation presented in this paper. Further, the following example illustrates another previously discussed [5] difficulty concerning the choice of quantitative parameters accounting for environmental cues. While it is well accepted that the outcome of T-lymphocyte stimulation is strongly dependent on the lifetime and force dependence of the TCR-ligand bond, there is currently no comprehensive list of binding properties containing all a priori information required to predict the outcome of receptor engagement. A possible way to overcome this difficulty may be to measure simultaneously with sufficient temporal resolution receptor engagement, force generation and cell activation. Thus, recent experiments suggested that TCR-mediated activation might be triggered by a force of about 5 pN, and this might be generated by a loading rate of 1.5 pN/s generated by T cells [180]. This kind of evidence might provide guidelines for determining the parameters relevant to cell activation, e.g., by using as a key parameter the bond lifetime under 5 pN force after a few seconds of strengthening. More generally, it may be suggested that a current challenge may consist of combining multimodal experimental platforms aimed at monitoring well-chosen cellular models [105] and aforementioned omic and machine learning techniques. While the complexity of this task may seem formidable, it may be hoped that a better understanding of cell encoding processes might help reduce the complexity of datasets to a manageable size.

## Figures and Tables

**Table 1 ijms-24-02266-t001:** How cells see the world.

Environmental Cue	Cell Detection Mechanism	Main Signaling Trigger
Soluble or anchored ligands of membrane receptors	Receptor engagement	Conformational change
2D topography of encountered surfaces—density gradients	Sorting of membrane molecules in contact areas—haptotaxis	Clustering, synapse formation
3D topography—roughness	Membrane curvature	Curvature-sensing molecules
Force generation and viscoelasticity of neighboring structures	Membrane undulations, pulling and pushing movement	Mechanotransduction

Cells continually analyze their environment, including the extracellular medium, extracellular matrix and encountered cells, to take decisions. A classification of retrieved cues is suggested, together with the way they are perceived by cells and the main signaling mechanisms involved in information processing.

## Data Availability

There is no specific dataset used in this review.

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
