# Peer review of "Understanding How Cells Probe the World: A Preliminary Step towards Modeling Cell Behavior?"

_ijms, 2023, doi:10.3390/ijms24032266_

Round 1
Reviewer 1 Report
It is hard to judge this perspective - it is not one to which I would naturally gravitate toward despite the topic - mechanisms by which cells probe their environment - being of great interest. There are no figures to guide the reader, no equations to provide a quantitative basis for description, only an essay like verbal discussion. At that some of the concepts seem less than well thought out. As one example, there is a section on "force generation", but what do these two words mean when used together? It is sensible to describe cells as doing work against an applied force, or to talk about energy generation, the storage form of which is potential energy. But there is no "potential force". It is one of those hazy phrases that are well used but that no one bothers to explain. Similarly signal generation. One can, e.g., generate or produce free Ca++ ion, but "signal generation" seems one of those filler phrases used to avoid having to think deeply about the ideas one wants to convey. It is however possible that others may find this perspective article useful for providing a framework through which exploration of cell-environment interactions can be viewed.
Author Response
I thank the reviewer for his thorough remarks that emphasize important points:
1) I tried to avoid equations to make this perspective attractive to less theoretically inclined readers. A
counterpart of this choice was a need to ensure that verbal concepts were clear and accurate, and I did my best to achieve this goal. I fully agree that it is important to try and avoid using "hazy phrases".
2) Force generation: It is certainly an open choice to emphasize the concept of either force or energy to describe mechanical effects. I felt that it was more appropriate to consider forces for the following reasons:
(i) forces are well established physical objects and, as pointed out by a leading theoretical physicist many decades ago : "we have at least a qualitative notion of force wich we acquire quite directly through the feeling we experience when using our muscles" (Arnold Sommerfeld, lectures on theoretical physics, vol. 1).
(ii) forces are often produced, or generated, during cell interactions with solid surfaces. Important biological examples are the hydrodynamic force exerted on a molecular bond tethering a blood leukocyte to the endothelial wall, the pulling force produced by a cell that has just bound a receptor ligand, or the protrusive force displayed by a migrating cell encountering a solid surfaces.
(iii) many authors reported on the influence of forces on biologically important phenomena such as lifetime of biomolecular interactions, conformation and exposure of reactive sites by biological molecules or membrane organization.
Thus, while I agree that it would have been sensitive to describe cells as doing work, I think that it is equally valid to speak about force or energy generation, and the choice is a matter of taste.
3) Signal generation: this expression was carefully chosen as explained in ref. [117], that was quoted as soon as the term of signal generation was used in the present paper. The basis of this choice is as follows :
- As written in the 5th edition of a leading textbook, Molecular Biology of the cell, and as quoted in ref. [117], "cells detect and respond to countless signals and communication between cells in multicellular
organisms is mainly mediated by extracellular signal molecules"
- This allowed us to write in ref [117] that our purpose was "to dentify and precisely define the signals generated by cell interaction with the surrounding world"
- Note also that in the 1977 printing of the "Shorter Oxford Dictionary", the relevant meaning of signal is defined as "a sign or notice, perceptible by sight or hearing, given especially for the purpose of conveying warning, direction or information"
Thus, I think that a signal may be defined as a message generated by cell interaction with its environment, as suggested in our previous review. A cell may respond (or not) to a signal by generating a signaling cascade, including many events such as a calcium rise.
I fully agree that we should have discussed this point in the paper, and I added a sentence in the revised paper.
Reviewer 2 Report
The author presents an interesting and detailed take on the state of cell modelling in molecular biology today while suggesting what could be done new using computational techniques from ML. While I like the idea, I am apprehensive on how good/accurate the predictive models will be. Moreover, a good predictive model requires a good training dataset, which is something worth investigating in this context. Given that the author mentions that environmental cues can hold crucial information and generally dictate cell behavior, it will be worth thinking about creating an extensive training dataset taking into account experiments that can measure the influence of various environmental on the macro/microscopic dynamics of cell behavior. With this additional commentary by the author along these lines, I recommend that the paper is accepted.
Author Response
I thank the reviewer for his positive comment and very useful criticism. Indeed, I agree that the quality of modeling is highly dependent on the definition of comprehensive datasets including suitable quantitative parameters for description of a variety of extracellular signals and behavioral responses. This important comment was added in the "conclusion".
Reviewer 3 Report
The perspective article titled 'Understanding how cells probe the world: a preliminary step towards modeling cell behavior?' attempts to consolidate and present the current approaches and methods employed to decipher how cells sense microenvironments and how local environmental cues play a pivotal role in regulating cell behavior. It presents a unique bidirectional approach where the authors first present the current research strategies and modeling approaches employed to predict focused cell functions. The author further breaks down the environmental cues to construct a minimal list of factors essential for cells to make smart decisions. Overall, the perspective is really well written with profound pieces of discussion to highlight the currently employed approaches, the strength of each method, the shortcomings or caveats associated with interpreting results with each approach, and the possible future research that might be essential to better our understanding. The author is systematic in his approach and has presented exhaustive literature compiled from diverse areas of research in order to provide an unbiased perspective toward understanding cellular behavior and function. A few minor concerns are listed below to further improve the quality of this perspective article. These should be considered as suggestions and incorporated to better present current literature and understanding in the field.
1) Overall, the author presents recent literature when presenting the ideas and the commentary. However, in the section describing the environmental cues, some of the literature cited seem to be more classical. The issue with presenting classical literature is that the recent progress made in the field may conflict with the findings presented in the perspective article. Additionally, presenting the recent findings will help establish a better understanding for the reader. See some examples below:
a) For example, the author discusses how different T cell receptor ligands drive diverse outcomes and attribute the differences to the lifetime differences of TCR-ligand complexes. The literature referenced to explain these findings is old. A more recent literature survey can provide a deeper understanding and explanation on the different mechanisms of action.
2) While explaining bond lifetimes, the author focuses on a particular type of bond (catch bond) without ever highlighting the other types (ideal vs catch vs slip). All these types of bonds are expected to play a vital role, especially during integrin adhesion maturation and turnover. A more recent literature survey can help ascertain the importance of other bond types in regulating cellular behaviors as they play a critical role in focal adhesion and cadherin adhesion formation.
3) In the section on force generation, the cited literature is fairly classical and does not represent the current status of the field. Most cited literature while pioneering is more than 10 years old and the understanding of scientists in the field has substantially evolved to better define the force orders and minimal force requirements. Example: A recent paper which attempted to confront T-cells on gel-phase SLBs observed that T-cell activation requires only approximately 5 pN force, while the force loading rates on the TCR are on the order of 1.5 pN per second - https://www.nature.com/articles/s41467-021-22775-z
Another example is the explanation provided with regard to adhesion established by CHO cells. It is mentioned that CHO cell adhesion to integrin ligand RGD requires forces higher than 40 pN while Notch receptors can tether on 12 pN surfaces. The issue with this classic literature is that these experiments were performed in the absence of any physiological growth factors or serum which restricted the typical adhesion formation observed in a native environment. More recent literature has challenged these findings and suggested that the use of growth factors such as EGF can help tune the forces required for adhesion formation and the established thresholds seem to be much lower than suggested by the presented literature. See reference - https://journals.biologists.com/jcs/article/133/13/jcs238840/225082/EGFR-activation-attenuates-the-mechanical
The important point to note is citing recent relevant literature might help present the current understanding in the field. At the very least, I encourage the author to present a comparison of classical literature with current literature as that seems to be the author's motive to provide an unbiased perspective with a focus on identifying a limited set of features used by cells to make smart decisions.
Author Response
I thank the reviewer for his positive comments and important remarks. The purpose of this review was to suggest a minimal list of signals processed by cells to take decisions, but I fully agree that I did not provide an exhaustive and up-to-date list of values of quantitative parameters accounting for these signals. This would indeed be a formidable task. I thank the reviewer for the excellent examples he suggested, and a paragraph was added at the end of the manuscript to emphasize the points that he raised. I felt that this provided a more appropriate conclusion than the last paragraph of the first version of the manuscript, and I hope this will be deemed satisfactory.